# High Biodiversity Arises from the Analyses of Morphometric, Biochemical and Genetic Data in Ancient Olive Trees of South of Italy

**DOI:** 10.3390/plants8090297

**Published:** 2019-08-22

**Authors:** Nicola Criscuolo, Francesco Guarino, Claudia Angelini, Stefano Castiglione, Tonino Caruso, Angela Cicatelli

**Affiliations:** 1Department of Chemistry and Biology ‘A. Zambelli’, University of Salerno, 84084 Fisciano, Salerno, Italy; 2Institute for Applied Mathematics ‘Mauro Picone’, CNR, 80131 Naples, Italy

**Keywords:** diversity, fatty acids, microsatellites, multivariate statistics, *Olea europaea* L., R.

## Abstract

Morphometric, biochemical and genetic analyses were conducted on *Olea europaea* L. of Campania, an area of Southern Italy highly suited to the cultivation of olive trees and the production of extra virgin olive oil (EVOO). We aimed to characterize the distribution of morphological, biochemical and genetic diversity in this area and to develop a practical tool to aid traceability of oils. Phenotypes were characterized using morphometric data of drupes and leaves; biochemical and genetic diversity were assessed on the basis of the fatty acid composition of the EVOOs and with microsatellite markers, respectively. We provide an open-source tool as a novel R package titled ‘*OliveR*’, useful in performing multivariate data analysis using a point and click interactive approach. These analyses highlight a clear correlation among the morphological, biochemical and genetic profiles of samples with four collection sites, and confirm that Southern Italy represents a wide reservoir of phenotypic and genetic variability.

## 1. Introduction

The origin of *Olea europaea* L. remains the subject of much debate today, but it has been hypothesized that wild plants were already present on the island of Crete in 4000 BC. Despite this uncertainty, it is nevertheless clear that for centuries olive has assumed an economic, symbolic and socio-cultural value for the people living in the Mediterranean basin. According to recent studies, human-mediated domestication of olive began in the Anatolian areas about 6000 years ago [1,2] and, over time, humans preferentially selected individuals with particular organoleptic characteristics and standards of olive oils. Despite its utility as an important cultivated species, the morphological and biochemical variability present in the olive trees in this region of Southern Italy is poorly characterized. Italy, as well as many other Mediterranean countries and contiguous areas, hosts a large part of the diversity of the species as evidenced by the description of about 600 varieties [3]. This diversity in the Campania Region (South of Italy) is known to be particularly rich, thanks to the local traditions and farming custodians, and several varieties and ecotypes are recognized for their socio-economic and environmental value [4]. The diversity of Italian olive germplasm has been characterized through the use of simple sequence repeats (SSRs or microsatellites) [5,6,7]. These molecular markers perform well in olive and exhibit a very low frequency of null alleles [8]. In general, they are extremely useful for quantifying the degree of intra-specific genetic polymorphism and can be used to determine kinship relationships among individuals sharing the same geographic area [9].

The construction of a genetic diversity database of olive trees based on microsatellites is not only relevant for botanical purposes, but it has the potential to provide a valuable tool that will lead to the improvement of the quality of extra virgin olive oils (EVOOs) and for facilitating their traceability. In addition to SSRs, morphological descriptors are the most used markers for identification and authentication of cultivars in regional, national and olive collections [10].

The EVOO is a key element in the human diet, whose exportation to the world by the European Union countries is around 65% of the total produced in Europe, with an income that exceeds one and a half billion dollars per year (http://www.internationaloliveoil.org/). The physiologist Ancel Keys focused his epidemiological studies on the diet of people living in the Cilento area (in South Italy, the Campania Region, in the province of Salerno), a district showing a very low incidence of chronic diseases and high human longevity. The results led to the definition of the nutritional model (now UNESCO heritage) known as the ‘Mediterranean diet’. According to this model, besides the constant use of fish, cereals, and vegetables, there is the daily consumption of monounsaturated and polyunsaturated fatty acids from EVOOs, in total contrast with the Western diet, which favors foods rich in saturated fats [11]. Further studies showed that triglycerides containing unsaturated fatty acids, in particular, the omega-3 and omega-6 family, provide several health benefits: Firstly, to the cardio-circulatory system, secondly, a reduction of the risk of osteoporosis fracture [12], and finally, a prevention of breast cancer and an adjuvant in the treatment of rheumatoid arthritis [13].

From previous works carried out on selected mono-cultivar olive trees, it is known that the fatty acid composition of the EVOO is above all genotype-dependent [14] and less influenced by other factors such as environmental or pedo-climatic conditions and agricultural practices, or fruit transformation processes, harvest times, etc. In this work we focused on *O. europaea* diversity investigated at different levels: morphometric, biochemical and genetic, but this time without any prior information regarding the olive tree varieties (thus characterized by a defined genetic and phenotypic profile). We collected different samples from a range of collection sites in the Campania Region, measuring morphological, biochemical and genetic parameters, shaped not only by geography but also by artificial selection, and by the preservation of specific properties of different EVOOs over time. The data were analyzed with an integrated approach using *OliveR*, an open-source software we developed in R. Results highlighted a high diversity among samples that originate from a relatively small geographic area.

Overall, the aims of this study are as follows:to study biodiversity and differences among, morphological, biochemical and genetic parameters in samples of *O. europaea* collected in four different areas of the Campania Region;to collect and curate a dataset containing geographical, morphometric, biochemical and genetic information about different olive trees of Campania Region, an area noted for its production of high-quality EVOO;to provide a novel R package (*OliveR*), a statistical user-friendly and open-source software used to carry out several integrated multivariate analyses on such types of datasets (e.g., one-way ANOVA, PCA and cluster analysis);to estimate the relationship between the macro-descriptors (and therefore the morphometric, biochemical and genetic variables) used to discriminate the olive trees and, for the first time, interpret the results in terms of the geographic location of the collection sites, to understand if the environment plays an essential role in determining the phenotype and thus the production of EVOOs.

## 2. Materials and Methods

### 2.1. Sampling 

A total of 169 trees of *O. europaea*, distributed in four areas of the Campania Region (South of Italy) belonging to the Provinces of Avellino and Salerno, were sampled in order to conduct morphometric, biochemical and genetic analyses on leaves and drupes. Drupe and leaf samples collection was carried out in mid-October for three consecutive years (2013–2015).

For the evaluation of the genetic biodiversity present in the Campania Region, we focused mainly on olive trees that are ancient or very ancient (trees probably over 500–600 years) but are still used for oil production. The identification of collection sites was carried out by means of *QuantumGis* software (version 2.18) using the geographic file (.shp) related to the Agricultural Land Use Card (CUAS) of Campania, (available for free at https://sit2.regione.campania.it/content/download). We exported the polygons that describe these areas to Google Maps and we easily manually identified the ancient olive trees from the satellite images because they are not arranged with a precise spatial organization (for example in rows). The geographic coordinates of all olive trees were recorded and used in a GPS navigator to proceed with the sampling.

The sampling areas were selected because they were very different from each other in terms of landscape features, elevation (the altitude is between 20 m and 610 m a.s.l., Appendix A) and socio-cultural aspects. On the basis of these criteria, we identified the following four broad tree collection sites as described in the following. Proceeding from north to south, two collection sites, corresponding to 48 and 26 samples respectively, belong to the Avellino Province (Irpinia area—hereafter labeled as OIR) and to the area of the Salerno Province (Sele river—labeled as OSE), respectively. The remaining samples were collected in the Cilento hinterland (collection site labeled as CE and consisting in 62 trees), and near the coast of the Cilento area (collection site labeled as CM and consisting in 33 plants), as shown in Figure 1.

A large number of the sampled olive trees are aged between 200 and 400 years (Appendix A), as estimated using diameter breast height (DBH) through the regression model for the age estimation defined in the case of monumental olive trees [15]. Furthermore, it was estimated that some olive trees are older than 600 years. Nevertheless, a small percentage of younger olive trees (0.11 m < DBH < 0.25 m) was sampled, particularly in the CE collection site and in some city squares of the Avellino Province, to understand if younger specimens bring greater variability compared to the average fatty acid content of ancient olive trees belonging to the same area. Some of these olive trees play a purely ornamental role in a city context, while most of the ancient olive trees are still used for the production of oil and therefore they were found in the land of private farmers.

A photograph of each tree was taken, after which small branches with at least 40 healthy leaves, showing no signs of disease or insect damage, were harvested from different parts around the crown of each olive tree. The leaves for morphometric analysis were immediately detached from the branch and placed between two sheets of paper; each pair of sheets containing the samples was pressed between two wooden planks held together by ropes and clamps to preserve the original shape of the leaf. In contrast, the leaves intended for molecular analysis were inserted in 50 mL labeled Falcon and immediately submerged in liquid nitrogen and, once in the laboratory, stored in a deep freeze at −80 °C prior to DNA extraction to avoid degradation of the material. 

Drupes were collected from each plant (about 1.0 kg per tree) brought to the laboratory and kept at 4 °C until oil extraction and for morphometric analyses.

### 2.2. Morphometric Data Collection

The leaf area of at least 20 pressed leaves per tree was measured using a portable device, equipped with an electromagnetic sensor, LI-3000C (LI-COR BIOSCIENCES, Lincoln, Nebraska), which returns a measurement to an accuracy of ±2.0%. To measure the major axis of the drupes, 20 drupes per plant were placed on graph paper and an image was taken. The images were imported into the Adobe Photoshop software (trial version CC 2015.5.1) and, through the perspective focus function and millimeter reference based on the graph paper, the major axis was estimated (Appendix A). Finally, at least 20 olives per tree were weighed on a technical balance, with an accuracy of ±0.01 g.

The values of the leaf area, major axis and weight of individual drupes were averaged for each sampled tree to obtain a single value for each of the three morphometric parameters per olive tree.

### 2.3. Molecular Analysis of SSR Microsatellites

Genomic DNA was extracted from leaves using CTAB based protocol. The six loci used for PCR amplification were chosen on the basis of their high degree of polymorphism, and because they were useful for the discrimination of different genotypes present in the study region [14,16]. The six loci include the following four belonging to the UDO family: UDO6, UDO17, UDO36 and UDO39 [17] and the following two belonging to the GAPU family: GAPU59 and GAPU71B [18]. The PCR reaction mix was prepared according to the procedure described by Cicatelli [14], whilst the PCR conditions were set up in the ABI 2720 Thermal Cycler, as follows: denaturing phase of 3 min at 94 °C was defined, followed by 35 cycles (40 s for GAPU71B) of 1 min (45 s. for GAPU59) at 94 °C, 1 min. (45 s. for GAPU59) at the annealing temperature of 58 °C (55 °C for GAPU59) and 1 min and 30 s (1 min for GAPU59) at the elongation temperature of 72 °C; to these cycles, a final phase extension of 7 min followed at a temperature of 72 °C.

The PCR products were analyzed by the genomic sequencer with capillary electrophoresis using the ABI310 Genetic Analyzer (Applied Biosystem, Milan, Italy). Gene Mapper 4.0 software allowed to estimate the size of the fragments of the loci in bp using as reference the 500 ROX Size Standard marker (Applied Biosystem, Milan, Italy).

### 2.4. Fatty Acid Composition of Extra Virgin Olive Oils

About 50–80 olives per sampled tree (depending on the size of the drupes for the diverse samples) were selected to obtain a weight equivalent to at least 100 g. The drupes were pitted and the remaining exocarp and mesocarp were mechanically crushed. The pulp obtained was centrifuged for 10 min. at 14,000 rpm. Finally, the supernatant containing the EVOO triglycerides was transferred into a sterile Eppendorf tube. 

The methyl esters of fatty acids were prepared according to EU Regulation n° 796/2002, then through trans-esterification with the use of a methanol and a potassium hydroxide solution. The methyl esters of fatty acids were detected by gas-chromatographic run using the 5975 VL MSD with Triple Axis Detector 7890 GC System, equipped with an automatic sampler and BPX5-SGE capillary column (50 mm × 0.22 mm × 0.22 μm); the resulting chromatograms were analyzed with the GCMSD Data Analysis software. The relative abundance of 10 fatty acids in each sample was assessed.

### 2.5. Statistical Analysis of Data

The analysis of the morphometric, biochemical and genetic data was carried out almost exclusively using specific packages available in the statistical software *R* (R Core Team 2017) and incorporated in *OliveR*, a novel open software R package we have developed specifically for our studies purpose. 

### 2.6. Analysis of Genetic Data

The following genetic statistics, allelic frequency (f), the number of alleles (n) and the expected ($H_{E}$) and observed ($H_{O}$) heterozygosity were calculated using the GenAlEx software (version 6.503; [19]) and the frequency of the null allele for each locus was calculated using the Brookfield formula [20].

We used the R package Adegenet 2.1.0 [21] to estimate the probability of genetic identity ($P_{\mathrm{gen}}$) for each unique genotype among individuals and to test for Hardy–Weinberg equilibrium for each locus by calculating the exact p-value with 100,000 permutations. We used the same package to derive a contingency matrix showing the presence or absence of a certain allele at each locus for each individual present in the dataset. We used the Dice–Sørensen index to calculate genetic similarity, s, between pairs of individuals, on a binary basis of presence (1) and absence (0) of a given allele for each amplified locus, and therefore we used the relative genetic distance ($d\;=\;\sqrt{\;\left(1-s\right)}$). Such distance was used to construct a dendrogram using the Ward method [22] as the linkage function. This method has proved to be the most appropriate for identifying a structure present in the population.

We used STRUCTURE software (version 2.3.2) for defining the population structure through the use of a model-based clustering algorithm applied to SSR markers [23], and we compared the results with those obtained from hierarchical clustering. More precisely, once the number of clusters in which to structure the population (in this case from K = 1 to K = 10) was defined, the admixture model was chosen to assign each individual to the most probable sub-population from which it inherited part of its genome on the basis of its allelic frequency. For the data available, a 50,000 step burn in-period followed by 50,000 MCMC repetitions was selected. The most plausible partition was chosen by analyzing the results obtained through the Evanno method [24] implemented in the Structure Harvester online platform [25].

### 2.7. Analysis of Real Data: Morphometry and Fatty Acids Content

We initially analyzed the relationship between each morphometric variable and the UTM North coordinate with a correlation test (*n* = 169) based on the Spearman coefficient.

Then, in order to study the effect of the collection site on the morphometric variables we fitted a one-way ANOVA model for each variable where the collection sites are seen as treatment groups. In particular, after having observed that the assumptions of normality and homoscedasticity of the residuals were not satisfied for all variables of our dataset—using Shapiro–Wilk test [26] and Levene test [27], respectively (Appendix A)—we applied the non-parametric version of the one-way ANOVA given by the Kruskal–Wallis test that orders the observations in ranks to calculate the differences among groups [28]. After that, the significance of each pair-wise comparison was carried out through a multiple comparisons test described by Siegal and Castellan [29], a non-parametric post-hoc analysis, in which, for each variable, the collection sites were compared in pairs to determine the actual difference. The value of α = 0.05 was used in order to measure the significance of the findings.

In the same vein, the Kruskal–Wallis non-parametric one-way ANOVA model was also fitted to investigate the relationship among the biochemical variables, defined in the fatty acid profile of each tree, and the collection sites, including the differences in the ratio between mono-unsaturated fatty acids (MUFA) and poly-unsaturated fatty acids (PUFA). The same threshold α was used for assessing the significance.

Moreover, for the analysis of the fatty acid profile, we also performed a principal component analysis (PCA) to reduce the dimensionality of the dataset. The first two principal components (PCs) were then used as input for the cluster analysis. The gap statistic test [30] was used to derive the most suitable number of clusters to group the available observations, and subsequently, the PAM partition clustering algorithm was applied to the biochemical samples. The quality of the inferred clusters was assessed using the analysis of the silhouette coefficient (ASC; [31]).

Finally, we compared the fatty acid clustering, obtained using the PAM algorithm, with the geographical distribution of the samples based on their collection sites using the adjusted rand index (ARI), in order to obtain a measure of similarity between two different partitions [32].

### 2.8. Integrated Multivariate Data Analysis

For each macro-descriptor (geographic coordinates, morphometric parameters, fatty acid profiles and genotypes) we computed the corresponding sample distance matrix. 

In particular, we used the Euclidean distance for estimating the matrices relative to the geographic coordinates and the morphometric data (since the data are quantitative on the real scale), the Dice–Sørensen index for estimating the matrix relative to the genotypes (since the data are binary), and Canberra distance for the fatty acids (in order to assign a greater distance to the observations described by the lower values [33]). To this purpose, it is useful to exploit that Canberra distance is particularly useful for data expressed in the form of percentages in which it is important to evaluate minimum differences between units, such as the presence, even in low quantities, of a fatty acid that is completely absent in another sample and that can, therefore, be considered a bio-marker.

Finally, we used the four above mentioned distance-matrices to perform a pair-wise correlation test between them using the Mantel test [34] choosing 100,000 permutations and a minimum significance level of α = 0.05.

### 2.9. Software Implementation

In order to facilitate the exploration and the analysis of our data, we implemented *OliveR*, a novel R package that allows the entire above-mentioned analysis to be performed, as well as to produce the figures using a point-and-click approach. *OliveR* was implemented in the R language using the shiny library. The tables and plots were made interactive through the functions of the plotly package.

*OliveR* has been structured in two main modules (quantitative data and genetic data), which are activated depending on the type of file/data (.csv and .shp) chosen by the user. Analysis of morphometric and biochemical data was performed using the quantitative module, the analysis of SSRs using the genetic module. In particular, loading a file with continuous quantitative variables, *OliveR* activates the layout shown in Appendix A. The navigation bar allows the user to query the dataset, perform several basic plots, compute the one-way ANOVA and calculate the principal components. It is also possible to perform a partitioning cluster analysis, as well as use appropriate tools to diagnose the analyzes carried out (e.g., through an ASC plot). By uploading a file containing genetic data, such as data relating to the size in bp of the alleles of particular loci, *OliveR* displays a layout as shown in Appendix A. In this case, the user can perform the Mantel test between different distance matrices, generate distance matrices with geometric and binary (similarity coefficients) methods to perform a hierarchical cluster analysis or visualize the genetic distance among samples through a heatmap. For both modules, by specifying the path to the folder containing the shapefiles, it is possible to represent the results of the statistical analyses on a geographical basis.

## 3. Results and Discussion

In order to achieve the main aims of the study, morphometric data on plant material were collected, six highly polymorphic SSR loci were PCR-amplified to infer the genetic biodiversity level. Furthermore, EVOO from the drupes of a single olive tree was purified and the fatty acid composition also determined. Finally, all the data produced were integrated and statistically analyzed.

### 3.1. Morphometric Parameters

The correlation test (*n* = 169) returned statistically significant values (*p*-value < 0.001) regarding the correlation of the morphometric variables with the geographic coordinate UTM North; this correlation was positive for the size of the olives (weight of olives: *r* = 0.693, major axis of the olives: *r* = 0.436 and negative for the leaf area *r* = −0.375, given that the leaves with the largest dimensions belong to the most southerly (CM) collection site.

The analyses of the variance for each morphometric parameter with respect to the collection sites carried out using the Kruskal–Wallis test (Appendix A, boxplot in Appendix A) showed statistically significant results further investigated through the multiple comparisons post-hoc test (Table 1, right side). The analyses of the morphometric parameters highlighted that there are morphological differences increase with geographic distance between sampling sites: e.g., among samples of CM (most southerly collection site) and OIR (most northerly collection site, Appendix A), while there are no substantial morphological differences between the CE and OSE collection sites which are located in much closer proximity to each other. The morphometric diversity assayed through morphometric features may, therefore, be useful to discriminate certain olive tree genotypes grown in situ, a finding that agrees with results of other studies carried out in South of Italy and in particular in Calabria and Sicily Regions [35]; since morphological characters are more liable to be affected by environmental factors [36], this suggested to consider them in association with biochemical and genetic traits for an accurate classification and identification of a variety.

### 3.2. Genetic Diversity and Population Structure

Indexes of genetic diversity based on the results of microsatellite analysis of the samples are presented in Table 2. A total number of 64 alleles were detected across the six loci used in the study and all loci were polymorphic. The mean number of alleles per locus was 10.7 and ranged between 9 for GAPU71B to 13 for UDO6. There was a zero probability of the existence of a null allele in all loci with the exception of locus UDO39 which had a non-zero probability of 23%. With the exception of the UDO39 locus, where $H_{O}$ is equal to 0.289, values of $H_{O}$ were always higher than 0.79 indicating high genetic diversity that is even higher than values found previously for other sular olive trees in the Mediterranean basin [37]. 

The dendrogram (Figure 2) clearly defines three distinct clusters grouped almost completely according to the collection site, although no a priori information was provided. Cluster 1 groups the 95.8% of the OIR samples, while Cluster 3 groups the 94% of the CM samples (sample P8 shows missing values in 4 loci out of 6 and for this reason it was inserted in cluster 1). Cluster 2 includes all the samples of the CE collection site and a sub-cluster including the 57.7% of the olive trees sampled in the OSE collection site. Another small part of the OSE samples is distributed within Cluster 1 and 2.

The hierarchical clustering of the six microsatellite-loci allowed the discrimination of 76 unique genotypes out of 169 (45%). This confirmed the finding of the previous study that Southern Italy is an important reservoir of olive germplasm biodiversity [14], whose sampling also included cultivars of Campania and Sicily, and detected 100 unique genotypes in the 136 olive trees analyzed. For each genotype the $P_{gen}$ was calculated, and it ranges from $4.6\times 10^{-2}$ to $6.2\times 10^{-6}$ for the members of CE, from $1.7\times 10^{-2}$ to $1.5\times 10^{-9}$ for CM, from $3.9\times 10^{-2}$ to $3.9\times 10^{-13}$ for OIR, and finally from $2.9\times 10^{-3}$ to $1.8\times 10^{-9}$ in the case of OSE collection site. These results indicated a very low probability of an allelic profile identical to another one occurring via sexual reproduction and can, therefore, be considered to be the product of vegetative propagation of a single clone. In particular, within the OIR collection site, there is just one genotype represented by at least four clones (referred to as genets).

Finally, in the dendrogram in Figure 2, eight different genotypes that include at least four clones were identified (list of samples belonging to these genotypes in Appendix A); in particular, the highest number of clones (18) was found in the CM group. It is interesting to note how some genets were found in different Municipalities (e.g., P2 and CT9 of CM collection site or SALV1 and BUC10 of CE collection site) belonging to the same collection area; the reason is attributable to the frequent exchanges of cuttings among farmers, living in close areas, due to the excellent EVOO quality produced (Appendix A).

Regarding the genotypic differences, allelic variations (in the order of two or three base pairs) were found among the CE and CM samples, and are probably due to somatic mutations that occur over time just afterwards vegetative propagation [17], since in this case the DNA is transmitted without any type of meiotic recombination, and therefore they could be considered molecular variants [10]. Although some members from the OIR collection site show a lower genetic similarity, they still have a common kinship, a sign that the SSRs are extremely selective molecular markers; in fact, even if the SSRs are available in limited number, good discrimination among different genotypes can be easily achieved. It was demonstrated that using just five SSRs, two cultivars, “Oblonga” and “Frantoio”, shared a very similar genotype and could, therefore, be considered as a unique variety [38].

The population analysis confirmed the results obtained previously with the hierarchical cluster analysis (Figure 2). Using 10 independent iterations for each possible number of clusters from 1 to 10, the mean value of the log-likelihood function used to estimate the best partition reached the maximum result at a number of clusters equal to 3, with values of ΔK = 89.36. The results of the analysis were extrapolated from the software and imported into R to generate the genetic admixture (q) plot keeping each labeled individual sorted by collection site. Again, the samples of the CE and the CM collection sites are included in clearly distinct genetic clusters, as well as those of the OIR collection site, where, however, a small number of individuals with a mixed cluster membership of clusters 2 and 3 is included, as also reported in the dendrogram. Also, in the STRUCTURE analysis, the samples belonging to the OSE collection site show a greater genetic similarity to those of the CE collection site.

The SSR molecular analysis and the two clustering methods not only allow detecting high biodiversity in a very restricted geographic area but also provide similar results. The maximum value of the “q” returned by the analysis with STRUCTURE for each sample was used to define the memberships. These memberships were compared to that one derived from the dendrogram through the use of ARI, which shows a value equal to 0.761, indicating excellent agreement between the two clustering methods.

### 3.3. Olive Oil Fatty Acids Composition 

Gas chromatographic analysis of oils allowed ten different fatty acids to be clearly identified. Some of these are common and occur in all the analyzed samples, while others are unique to the oils of samples sourced from particular geographical areas. Table 1 shows the average relative abundance of each fatty acid for each collection site. These results showed that it is possible to identify the sampling site origin of plants from the fatty acid composition of their oils (as also found in the boxplot analysis in Appendix A). For example, the percentage of oleic acid varies in a range between 70% and 80%, with the highest percentage in the oils extracted from OIR and OSE samples. However, other fatty acids such as heptadecenoic, t-octadecenoic, eicosanoic, eicosenoic and behenic, are not always present in every olive oil extracted by the samples of all collection sites.

The results of the Kruskal–Wallis test applied to each fatty acid (Appendix A) suggested that the biochemical profile of the EVOOs strictly depends on the geographic location of the sampling site (*p*-value of the $\chi^{2}$ statistic <0.001). The post-hoc nonparametric test (Table 1 right side) for the analysis of variance showed that the major differences in the biochemical profile of the olive oil are attributed to the samples belonging to CM and OIR collection sites (significant difference for 9 out of the 10 fatty acids), as is also the case for morphometric variables. Moreover, CE collection site deviates from the OIR and CM for the content of 7 of the 10 fatty acids. Similar fatty acid concentrations were found for the members of the OIR and OSE collection sites (8 out of 10).

In addition, Table 1 also reports the reference limits of the EU Delegated Regulation 2015/830 that are used to classify an oil as an EVOO; it is notable that a large part of our samples meets the composition criteria to qualify as EVOO. In this table, for every collection site is also reported the corresponding MUFA/PUFA ratio. An essential characteristic of this type of oil is the high value of the MUFA/PUFA ratio, which, for each collection site, is higher than 8.0, confirming that the EVOOs are a rich source of monounsaturated fatty acids. The Kruskal–Wallis test and its post-hoc analysis were then applied also to the MUFA/PUFA ratio based on the collection site. A significative difference was found for almost every pairwise comparison (Table 1 and Appendix A), which is in line with the difference found between the singular fatty acids. The MUFA/PUFA ratio, in association with a specific diet, has been linked to a significant reduction in the risk of cardiovascular diseases [39]. The high MUFA/PUFA ratio can contribute to a diet rich in monounsaturated fatty acids which are thought to protect low-density lipoprotein (LDL) particles from oxidation, which can lead to the formation of atherosclerotic plaques. Regarding polyunsaturated fatty acids, the CM and OIR samples show the highest concentration of linoleic (omega-6) and linolenic (omega-3) acid, respectively.

The biochemical data were subjected to PCA in order to highlight the meta-variables that were useful for the discrimination of samples. The first two PC axes explain a total of about 75% of the variation (relative weights of PC1 and PC2 in Appendix A) and allow the samples to be clustered into at least three macro-groups (Figure 3). We observed that one of the groups is dominated by CE samples, another by CM samples, while the last group mostly contains samples from OIR and OSE collection sites. Such results confirmed, as also showed by the Kruskal–Wallis test, that the biochemical profile allows the origin of samples to be clearly discriminated, and the PCA allows fatty acids composition of each sample to be characterized.

It is notable that higher values of palmitoleic and t-octadecenoic acids are typical of olive trees sampled from the CE collection site, while higher values of linoleic, eicosanoic, palmitic, behenic and heptadecenoic are typical of samples from the CM collection site. Members of the OIR and OSE collection sites occur in a similar position in the PCA and are characterized by higher values of oleic, linoleic and eicosenoic acid even if there are some olive oils extracted from trees of the Avellino Province with a different biochemical profile from that of other oils obtained from plants of the same Province. This is the case of OIR2, OIR4, OIR73 and OIR74 samples probably imported from other geographic sites and used for ornamental purposes.

To confirm the results illustrated in Figure 3, the first two PC axes were used to conduct a cluster analysis with the PAM method. As shown in Figure 4a, the point at which the gap statistic levels off indicates that the most likely number of biochemical clusters is three. This is confirmed by the analysis of the ASC (Figure 4b), which shows a mean value of 0.73, again indicating that the most likely number of clusters is three. The location of members of the three clusters is shown in Figure 4c. Members of biochemical Cluster 1 were largely restricted to sampling site CE and members of Cluster 2 were mostly confined to sampling site CM. However, members of Cluster 3 occurred in both OSE and OIR collection site. Nevertheless, the value of ARI for the comparison among the cluster partition (three clusters) and that of the collection sites (four sites) is equal to 0.739. Values close to 1.0 indicate a good overlap between the two partitions.

In particular, 100% of the CE collection site samples are included in cluster number 1, 97% of the CM collection site samples in cluster number 2, while 91.7% of the OIR and 100% of OSE collection sites samples share cluster number 3 (Appendix A). As hypothesized on the basis of the biochemical PCA, few members of the northwest Salerno Province are classified in a rather heterogeneous way, probably due to their uncertain provenance; however, the results achieved only through the use of the biochemical profiles are extremely useful for differentiating between these samples. Our results demonstrated that it is possible, on the basis of fatty acid content, to identify the source location of individual samples taken from across a relatively small geographic area such as that of two Provinces. This is despite the fact that the two Provinces are both extremely heterogeneous in terms of pedo-climatic conditions and socio-cultural traditions. Moreover, since some fatty acids were found exclusively at a single collection site, they offer the potential to use them as bio-markers for a more detailed classification of the commercially important cultivars at the national level. Indeed, some ecotypes still used for the production of EVOOs could show superior oil composition to widely used recognized cultivars [40], and therefore this type of analysis could prove extremely useful for the assignment of the PDO certification mark to this type of product.

### 3.4. Relationship between the Macro-Descriptors

An initial analysis to partition the selected samples was carried out using the morphometric and biochemical traits alone. The morphometric and biochemical parameters do not group the sampling sites in exactly the same way. Although the characterization through the use of fatty acids seems to be more effective in recognizing the origin of specimens on a geographic basis, the morphometric characters of leaf and fruit size did clearly differentiate material from CM and OIR collection sites into two discrete groups. 

Morphometric and biochemical parameters were then used simultaneously in a multivariate study to improve the differentiation of the groups in relation to the collection site. In fact, the results of the PCA and cluster analysis, performed on a dataset with both types of phenotypic variables, show that it is possible to obtain better separation of the clusters, as shown in Appendix A. Moreover, these morphometric parameters better separate the OIR samples from the OSE ones. In this case, the best clustering was obtained with four clusters (as many as the collection sites, instead of three) since the ARI value increases up to 0.768 (Appendix A). Next, we quantified the degree of correlation, *r,* among morphometric, biochemical and genetic traits, in order to understand if the geography, and therefore the pedo-climatic and environmental conditions have over time played a dominant role in determining the various properties of olive varieties found in Italy today. Furthermore, the geographic distance could also reflect reproductive isolation due to artificial selection, operated by farmers in the past, who selected the best performing varieties for that pedo-climatic conditions, and from which, through vegetative reproduction, have been spread in different local ranges. Table 3 shows the correlation coefficient, *r*, and the results of the Mantel test carried out between the distance matrices of each macro-descriptor (including the geographical coordinates), both for the complete dataset and for the genets (belonging to the eight genotypes identified above) dataset. 

In general, the results were characterized by a very high level of significance (*p*-value simulated <0.0001) and showed a positive correlation for all cells listed in Table 3. Considering the dataset with all the samples (Table 3A), the correlation between the morphometric, biochemical and genetic parameters was in the range 0.169–0.485. Therefore, the correlation between these traits is significant but not particularly strong. This is in contrast to the findings of Rotondi et al. [41] who reported a strong relationship between the genotype and the composition in fatty acid of the olive oil but which focused the attention specifically on nine pre-selected monovarietal oils. On the other hand, in our results, there is a marked evidence for isolation by distance as there is a strong correlation (*r* = 0.641) between the genetic and geographical distance. This agrees with the results of a study of olives carried out in six different geographical areas of Africa using both microsatellites and chloroplastic DNA as molecular markers [42]. There is also a greater correlation between the biochemical composition and geographical distance (*r* = 0.743), whereas that for morphometric traits and geographic distance is much weaker (*r* = 0.303).

On the basis of this analysis, we conclude that individuals with certain genetic and phenotypic parameters have been subjected to both natural selection so that they suit environments with particular pedo-climatic characteristics and to human-mediated selection to produce plants with oil of the desired composition. Since olive cultivation is carried out for the production of EVOO, in each of the considered areas, tree selection was certainly made in order to preserve plants that are highly resistant to cold or to pests as well as their ability to produce an EVOO with organoleptic characteristics that are unique to the local area which reflect the gastronomic and cultural traditions and of course agriculture practices of the Campania region. In many cases this was achieved by vegetative propagation of desirable genotypes; the proof of men’s action is shown in Table 3B, where the correlation between SSR profile and geographic site calculated for the clone dataset reaches a value of 0.867. For instance, it is known that in the case of the OSE samples (Province of Avellino and Salerno close to Sele river) there is a long tradition of tree coppicing in order to bring about rejuvenation so that individuals with the desired traits can be propagated and maintained over long periods of time. Mass propagation of this type may have led to an increase in certain allele frequencies within the sub-populations of Campania, and it is likely that, for the considered loci, they are not in Hardy–Weinberg equilibrium. In our study, this is supported by the very strong correlation between biochemical composition and geographical distance.

The strong correlation of the biochemical matrix with the geographic origin of olive trees could, however, be due to some differences in agricultural practices, such as irrigation, that can influence the characteristics of the final product, as the proportion of PUFA, the content of sterols and other volatile components, and also bitterness and acrid taste [43]. These practices are arbitrarily chosen by the farmers on the basis of established production standards handed down by local culture and gastronomic traditions. Different cultivation methods could explain why some OSE samples, which are located halfway between the Irpinia (Province of Avellino) and the Cilento (Province of Salerno), show a similar biochemical profile closer to the OIR samples, but a genetic profile more similar to that of CE. To explain this, we have to consider the fact that the microsatellite loci are neutral markers; therefore, they are not involved in the phenotype determination. For this reason, an integrated approach to the characterization of olive oil is required.

The correlation between the different distance matrices recalculated for the subset of data containing only the samples belonging to the eight different genotypes which include at least four clones (Table 3B) increases for each comparison, in particular for the relation of the genotype with the morphometric parameters of drupes and leaves, although they can be partially influenced by environmental characteristics such as the different seasonal climatic conditions. On the other hand, the relationship between the genetic data and the fatty acid composition of olive oil is strengthened. This increase in the r value confirmed how the genetic diversity, due firstly to environmental and subsequently to human-mediated selection, has been translated over time into a phenotypic biodiversity that is still maintained for the production of EVOOs with peculiar characteristics, even if the territory of Campania region is very heterogeneous in terms of pedo-climatic conditions, despite the small area studied. Moreover, having preserved the EVOO quality standards for a long time (starting from very ancient ancestors) even the youngest olive trees have not shown biochemical characteristics substantially different from the ancient ones.

### 3.5. Software Availability and Data Repository

The structure of *OliveR* is shown in the Appendix A. *OliveR* was used to carry out almost all the data analyses presented here and to produce the main figures of this study. *OliveR* can be used for the analysis of any similar dataset. The advantages of using *OliveR* are the following: (i) thanks to a user interface consisting of well-defined sections, drop-down menus and intuitive command inputs, users do not require any knowledge of a programming language; (ii) it is a fast point and click package that can be used to explore and analyze different kinds of data quickly and easily, and therefore it increases the number of explorations that can be carried out; and finally (iii) the facility to generate interactive plots that make it easy to extract information on groups of statistical units or individual samples; in fact, by simply passing the cursor on an element of the graph it is possible to immediately know the identifier of that specific sample, the group to which it belongs, or other relevant information.

*OliveR* is freely available on the GitHub platform (https://github.com/nicocriscuolo/OliveR) with the relevant instructions on how to install it in RStudio; an example of the analysis with a video-tutorial is also available. Moreover, all data collected in this study are available at https://github.com/nicocriscuolo/OliveR/tree/master/inst/CSV_data as text files in .csv format. The data are ready to be directly imported into *OliveR*. 

## 4. Conclusions

This work focused on 169 olive (*O. europaea*) trees of Campania Region (South Italy) and investigated the relationships among the most suitable parameters useful to describe their diversity. The integrated approach of morphometric, genetic and biochemical data highlighted a high biodiversity of olive trees in the four different collection sites, which are located in a relatively small area in the South of Italy. Our work has allowed us to build a dataset containing different kinds of data, and the development and use of the interactive statistical software *OliveR* have also made our work of integration much faster and more accessible, allowing us to obtain interesting results in a shorter time. This integrated approach allowed us to better discriminate some olive trees (CE and OSE), sampled in two different Campania provinces (Avellino and Salerno), that were genetically clustered together, whilst they were well separated on the basis of EVOO fatty acids composition. These results suggest that, although the genetic strongly characterized the quality of EVOO, the geographical position, with all socio-economic, tradition, agriculture practices and pedo-climatic implications have to be considered in such biodiversity studies and for an accurate selection of the cultivars. Moreover, the integrated approach suitable in *OliveR* allowed us to identify proper markers for the traceability of EVOOs of the studied areas. The implementation of datasets with data coming from other provinces, regions or countries might allow the identification of markers suitable in the PDOs definition and for tracing their quality.

## Figures and Tables

**Figure 1 plants-08-00297-f001:**
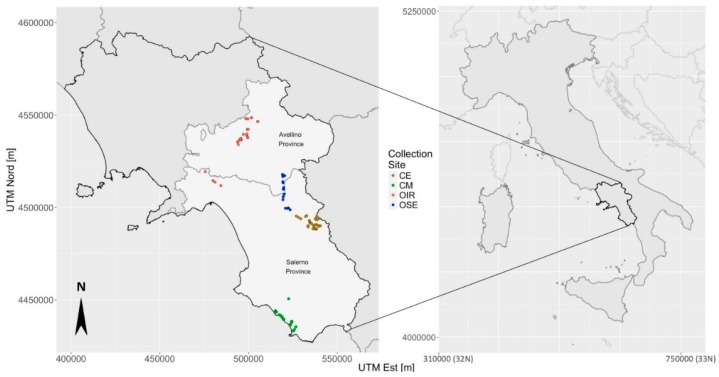
Geographic distribution of the 169 olive trees in Campania Region and corresponding tree collection sites [CE: Cilento hinterland (brown color), CM: Cilento coastal area (green color), OIR: olives of the Irpinia area (red color), OSE: olives along the Sele river (blue color)]. The projection system used to generate geographic maps is the Universal Transverse Mercator with the distances measured in meters (Appendix A).

**Figure 2 plants-08-00297-f002:**
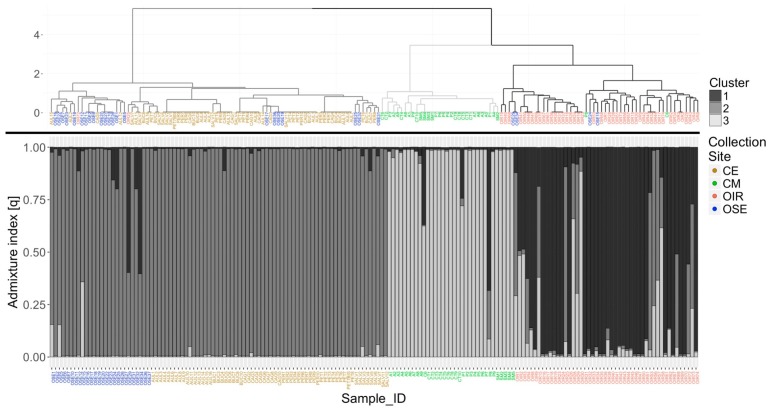
Comparison between the dendrogram calculated with the Ward method on the genetic distance among samples, based on the similarity index of Dice–Sørensen (above), and the clustering obtained through the population analysis conducted with the STRUCTURE software [CE: Cilento hinterland (brown color), CM: Cilento coastal area (green color), OIR: olives of the Irpinia area (red color), OSE: olives along the Sele river (blue color)].

**Figure 3 plants-08-00297-f003:**
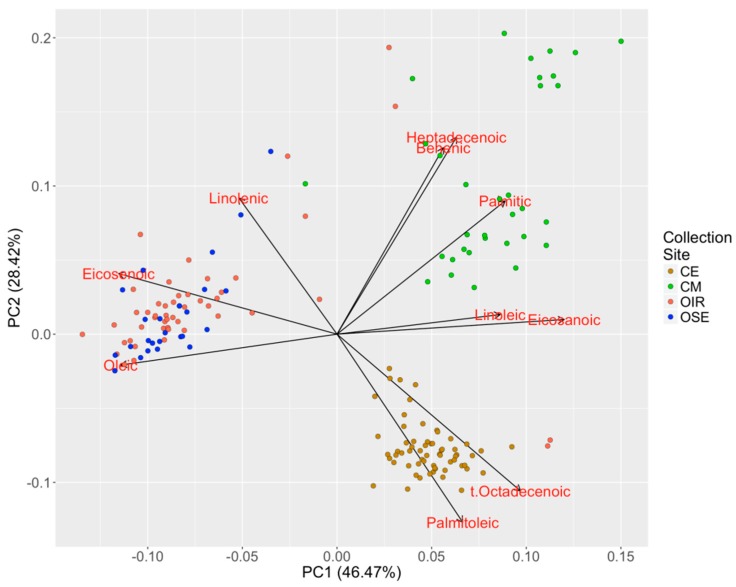
Biplot of the first two principal components (PCs) calculated on the fatty acid dataset. The scaled values of the loadings calculated on the basis of the original variables are shown in red.

**Figure 4 plants-08-00297-f004:**
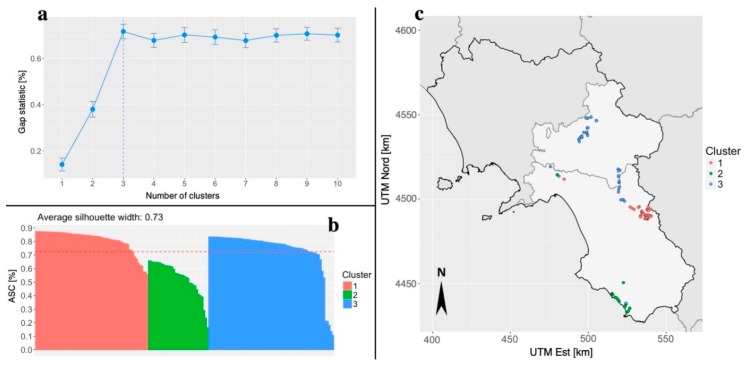
Panel plot for the cluster analysis performed on the first two PCs calculated on the fatty acids dataset. It is reported the number of clusters that maximize the value of the gap statistic (**a**), the individual silhouette of each cluster the average silhouette as a measure of cluster internal accuracy (**b**) and the cluster membership of each sampled tree reported on the geographic basis (**c**). Cluster n° 1 (red color) contains all samples of the CE collection site and samples OIR73 and OIR74; cluster number 2 (green color) contains 32 out of 33 samples of the CM collection site and samples OIR2 and OIR4; cluster number 3 (blue color) contains all samples of the OSE collection site, 44 out of 48 samples of the OIR collection site and sample SM5.

**Table 1 plants-08-00297-t001:** Mean values and standard deviation of morphometric parameters and fatty acids concentration divided for the four collection sites (left) and results of the Kruskal–Wallis post-hoc test (right) with multiple comparison test for every measured parameter. Significant differences between pairs of groups are represented by the T (TRUE); non-significant differences are represented by the F (False).

Mean Values and Standard Deviation of Morphometric Parameters and Fatty Acids						Kruskal-Wallis Post-hoc Test					
	Collection Site					Pairwise Comparison between Collection Sites					
	CE	CM	OIR	OSE		CE-CM	CE-OIR	CE-OSE	CM-OIR	CM-OSE	OIR-OSE
Morphometric Parameters											
Olive Weight [g]	1.63 (0.37)	1.13 (0.26)	2.12 (0.45)	1.73 (0.23)		T	T	F	T	T	F
Major Olive Axis [cm]	1.81 (0.22)	1.71 (0.18)	2.03 (0.26)	1.82 (0.21)		F	T	F	T	F	T
Leaf Area [${\mathrm{cm}}^{2}$]	4.71 (1.06)	7.14 (1.63)	4.72 (1.24)	5.09 (1.32)		T	F	F	T	T	F
Fatty Acids [%]	-	-	-	-	Reference Values of EVOO (%)	-	-	-	-	-	-
C16:0 (Palmitic)	12.69 (0.71)	16.35 (2.93)	12.51 (1.97)	10.63 (1.2)	-	T	F	T	T	T	T
C16:1 (Palmitoleic)	2.23 (0.44)	0.9 (0.68)	1.03 (0.57)	0.54 (0.18)	0.3–3.5	T	T	T	F	F	T
C17:1 (Heptadecenoic)	0 (0)	0.33 (0.09)	0.04 (0.11)	0.01 (0.02)	≤0.3	T	T	F	T	T	F
C18:1 (Oleic)	72.86 (1.79)	70.72 (3.36)	76.94 (5.14)	80.01 (3.12)	55.0–83.0	F	T	T	T	T	F
C18:1t (t-Octadecenoic)	3.84 (0.53)	1.49 (0.53)	0.27 (1.06)	0 (0)	-	T	T	T	T	T	F
C18:2 (Linoleic)	7.47 (1.22)	8.06 (1.66)	5.06 (3.46)	6.29 (1.68)	2.5–21.0	F	T	F	T	T	F
C18:3 (Linolenic)	0.47 (0.12)	0.68 (0.22)	1 (0.38)	0.74 (0.2)	≤1.0	T	T	T	T	F	F
C20:0 (Eicosanoic)	0.45 (0.07)	0.63 (0.15)	0.02 (0.09)	0 (0)	≤0.6	T	T	T	T	T	F
C20:1 (Eicosenoic)	0 (0)	0 (0)	0.59 (0.2)	0.61 (0.09)	≤0.4	F	T	T	T	T	F
C22:0 (Behenic)	0 (0)	0.09 (0.03)	0 (0.01)	0.02 (0.05)	≤0.2	T	F	F	T	T	F
MUFA/PUFA	10.23 (1.96)	8.72 (1.82)	17.37 (8.42)	12.40 (3.61)	-	T	T	F	T	T	F

**Table 2 plants-08-00297-t002:** Genetic indexes calculated with the GenAlex software. Allelic frequency (f), number of amplified DNA samples (N), number of different alleles ($N_{a}$), effective number of alleles ($N_{e}$), frequency of the null allele (r), observed ($H_{o}$) and expected ($H_{e}$) heterozygosity, weighed expected heterozygosity ($uH_{e}$) and probability of the exact Hardy–Weinberg test (p -value < 0.001, S: significant, NS: not significant).

	UDO36	*f*	UDO6	*f*	UDO17	*f*	GAPU59	*f*	GAPU71B	*f*	UDO39	*f*
	138	0.012	144	0.012	144	0.003	204	0.009	116	0.009	104	0.012
	140	0.190	146	0.373	150	0.003	206	0.259	118	0.015	144	0.093
	142	0.075	160	0.088	152	0.155	208	0.003	120	0.479	164	0.003
	144	0.461	162	0.003	154	0.211	210	0.449	122	0.006	170	0.503
	148	0.003	164	0.003	156	0.082	212	0.006	124	0.015	174	0.012
	150	0.006	166	0.006	158	0.007	214	0.003	126	0.140	176	0.166
	152	0.235	168	0.158	160	0.118	216	0.098	128	0.003	178	0.024
	154	0.006	170	0.055	162	0.280	218	0.003	140	0.330	180	0.160
	160	0.006	172	0.252	168	0.125	220	0.164	144	0.003	186	0.006
	162	0.003	178	0.003	170	0.016	224	0.006			188	0.018
	164	0.003	180	0.042							190	0.003
			188	0.003								
			190	0.003								
Index												
N	166		165		152		168		168		166	
$N_{a}$	11		13		10		10		9		11	
***N_e_***	3.231		4.171		5.460		3.272		2.786		3.165	
$H_{o}$	0.910		0.861		0.796		0.792		0.946		0.289	
$H_{e}$	0.690		0.760		0.817		0.694		0.641		0.684	
$uH_{e}$	0.693		0.763		0.820		0.696		0.643		0.686	
***r***	0		0		0.01		0		0		0.23	
***H*−*W* test**	*S*		*S*		*S*		*S*		*S*		*S*	

**Table 3 plants-08-00297-t003:** Results of the Mantel test (*n* = 169) carried out between pairs of four different distance matrices defined for macro-descriptors (for every comparison we report the correlation coefficient *r*, simulated *p*-value < 0.001 ***).

Mantel Test, *r* ***	A. Distance Matrices—All Samples		
	Morphometric Parameters	Fatty Acids	SSR
Geographic Coordinates	0.303	0.734	0.641
Morphometric Parameters	-	0.169	0.246
Fatty Acids	-	-	0.485
	**B. Distance Matrices—Genets Dataset**		
	**Morphometric Parameters**	**Fatty Acids**	**SSR**
Geographic Coordinates	0.365	0.827	0.867
Morphometric Parameters	-	0.196	0.438
Fatty Acids	-	-	0.620

## Data Availability

All data collected in this study are available at https://github.com/nicocriscuolo/OliveR/tree/master/inst/CSV_data as text files in .csv format. The data are ready to be directly imported into *OliveR*. The data concerning other samples information (trees photographs, trunk circumference and data related to the morphometric parameters, that were subsequently averaged to have a single measurement per sample), are available on request.

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
