# Peer review of "High Biodiversity Arises from the Analyses of Morphometric, Biochemical and Genetic Data in Ancient Olive Trees of South of Italy"

_plants, 2019, doi:10.3390/plants8090297_

Round 1
Reviewer 1 Report
Plants Peer Review
Excellent methodological approach for botanical varieties. Very well presented and described, specially the diver’s software packages. However, the statistical design and analysis are not well suited as the cultivar and environmental factors are confounded in the analysis. The R olive package would represent a valuable contribution for botanical varieties, but not for cultivars if they are not previously identified (true to type.
The main hypothesis of this work is the determinism of geographic site in the EVOO composition. However, the cultivar, a main factor in the EVOO composition, is not considered in the study. Cultivar is usually associated to the geography as stated in all olive cultivars catalogue in traditional olive growing countries. particularly in old plantations, as it is the case of tis work. Therefore, the correct identification of the cultivar must be compulsory to support this hypothesis.
Summarizing, the paper must be revised and resubmit previous identification of the cultivars corresponding to the 169 trees. and. Cultivars instead of trees have to be used for the analysis.
P1 l 4. b is missing in the authors names
P 1 l 26. Within geographic origin should be considered the cultivars as their geographic distribution in all Mediterranean countries is strongly associated to their area of selection where they have been vegetative propagated across time. See lines 28 and 62
P1 l 29 Ecotype is a misleading term in olive vegetative propagated cultivars as it may only refers to the G x E interaction. The term may have sense for wild and feral olives where natural sexual reproduction and adaptation to the environment are the major factors building up biodiversity.
PDO directive only considers one or few major cultivars of the geographic site.
P1 l 38.
Any country hosts large number of cultivars that usually differs from those of other countries. The diversity of cultivars is a common trait in any Mediterranean country. Major cultivars differ between close contiguous geographic areas in any olive producing country.
P1 l 40. The concept of ecotypes is confused in olive cultivars. Please clarify
P2 l 46. SSRs and UPOV morphological descriptors are currently the most used markers for identification and authentication of cultivars in regional, national and world collections of cultivars.
See for instance Trujillo et al. 2014. I agree with the use of SSRs for botanical studies but for PDO the authentication (true to type) of cultivars with both SSRs and UPOV morphological descriptor should be a prior step. Otherwise the terms geographic site end cultivars are confounded. See line 26.
P2 l 65. You do not provide “any prior information regarding the olive tree varieties”. As the cultivar is a major factor in PDO, true to type cultivars used for the EVOO must be a factor to be studied in this work. See notes in lines 38, 39 and 46.
P 2 l 88. Excellent methodological approach for botanical varieties. Very well presented and described, specially the diver’s software packages. Please clarify the concepts related to plant material: unique genotype, ecotype, genet, ramet, clones indicating their sources of variability
However, the statistical design and analysis are not well suited as the cultivar and environmental factors are confounded in the analysis. The R olive package would represent a valuable contribution for botanical varieties, but not for cultivars if they are not previously identified (true to type).
Again, the absence of the factor cultivar in the study is critical for the conclusions. At his regards the morphometry study is insufficient to identify the cultivars corresponding to the 169 sampled trees.
P5 l174. Phenotipic
P5 l 177. Morphometric data are insufficient to identify cultivars
P7 l 267-268. Which are the criteria considered by the authors for an accurate classification and identification of a variety or ecotype.
P7 l 279. Which are the effects of cultivar and environment of the geographic site in the MUFA/PUFA ratio
P9 l 325. Which are the cultivars associated to each cluster, site and sampled tree.?
P 10 l 338. See previous notes
P11 l 368. The authors should justify the proposed threshold of genetic similitude (75%) to discriminate unique genotypes-Here appear once more the confusion between botanical populations and cultivars. I suggest follow the criteria used by Trujillo et al. (2014) that allow discriminate more than 833 trees corresponding to 499 different accessions from 21 countries. The use of 33 SSRs and 11 morphological traits allows discriminate 323 different cultivars.
P 11 l 391. Me parecen variantes moleculares. ¿Tienes a mano la publicación de Kruskal Wallis (18)?
P13. Table 2. Confounded environment and cultivar in Geographic coordinates (Table 2)
P14 l 473. Hardy-Weinberg should not be used for the probably few cultivars of any PDO.
P15 l485. This is the reason why SSRs should be complemented with UPOV morphological markers to identify the cultivars
P15 l 498. The main hypothesis of this work is the determinism of geographic site in the EVOO composition. However, the cultivar, a main factor in the EVOO composition, is not considered in the study. Cultivar is usually associated to the geography as stated in all olive cultivars catalogue in traditional olive growing countries. particularly in old plantations, as it is the case of tis work. Therefore, the correct identification of the cultivar must be compulsory to support this hypothesis.
P15 l 537.
'Ravece', the only cultivar mentioned in the work, confers the typicity of the province of Avellino EVOO as the authors stated. It is a significant data that support my main objection to be the environmental site the only geographic factor responsible for the EVOO composition. Cultivar must be considered in this study.
Reviewer 2 Report
Title: High Biodiversity Arises from the Analyses of 2 Morphometric, Biochemical and Genetic Data in Ancient 3 Olive Trees of South of Italy
The subject of this manuscript falls within the general scope of Plants. The manuscript is an original contribution to the phenotypical and genetic diversity of Olea europaea in Italy and their combined analysis with clear useful applications. I would recommend the publication of this manuscript in Plant Ecology after major revisions.
Abstract
Line 14. Please, change ‘phenotypic’ to ‘morphological’, since you are also studying ‘biochemical’ phenotypes. Please, apply this orientation all through the text. Ex. Abstract Line 15; Introduction, Line 36.
Keywords should be in alphabetical order. Avoid words included in the title.
Highlights
Please, delete ‘Southern Italy is renowned for olive cultivation and EVOO production’. This is not a conclusion from your study.
Use the full words for ‘PDO’.
Introduction
Line 35. When did domestication started? Please, add the date.
Line 41. Please, add a reference to this statement after ‘value’.
Line 50. Please, add a reference to this statement after ‘year’.
Line 56. Please, add a reference to this statement after ‘fats’
Lines 70-74. Delete ‘Results highlighted a high diversity among samples that originate from a relatively small geographic area; moreover, the results of this analysis allow us to identify the provenance of our samples and to trace the origin of many types of EVOOs produced in Campania. This is very important especially for those EVOOs showing high values of selected fatty acids, essential for a balanced diet and prevention of many types of diseases’ from the Introduction.
Please, add your hypothesis at the end of the Introduction.
Materials and methods
Line 109. Add a table containing the geographical coordinates of each sampled tree and other interesting information, such as the province, altitude, estimated age, etc.) to supplementary materials.
Line 183. ‘non-parametric post-hoc analysis’. Which one? Man-Whitney U-test?
Results and discussion
In general, Results should be written in past tense.
Lines 253-257 and thought the text. Please, add the ‘n’ for each correlation analysis.
Line 258-259 and thought the text.. You do not need to say again the test you applied since you already say it in M&M section.
Line 308. Please, add a table to supplementary material showing the relative weights of every fatty acid on each principal factor (PC1 and PC2).
Lines 438-439. Not only ‘and therefore the pedo-climatic and environmental conditions’. Geographic distance may also reflected reproductive isolation. Please, comment on this aspect.
In general, I would be very conservative using the term ‘ecotype’ for a domesticated tree. Please, explain your use the first time you mention it in the text.
Are climatic changes (temperature, rainfall, etc.) derived from climate change changing the morphological and biochemical traits that you recoded? May be it would be interesting to comment on this in your Discussion?
Tables
Table 1. delete ‘. For the post-hoc test the level of significance is set at 5% for every pairwise comparison’ from the table heading.
Figures
Fig. 1 and 4. Add the meaning of labels (CE., CM, etc.) to the figure legend.
Fig. 4. Tree labels are too small (impossible to see). You could put numbers instead of labels and use these numbers also in the new suggested Table in M&M for supplementary material.
Round 2
Reviewer 1 Report
Dear corresponding author,
I appreciate the changes you have incorporated to the Ms. The changes have significantly improved and clarified the first version. Unfortunately, I believe that some extra changes are required to completely overcome the bias that the incomplete analysis presented in Table S5 represent. Also, to clarify some confusion in the terms used to distinguish between botanical varieties and cultivars. I propose the following changes to your consideration:
1) Related to Table S5, I believe you must modify it, namely the number of genotypes with clones. Why did you consider only the genotypes with more than four clones (designed ramets in the corresponding heading of the table)? I have recalculated Table S5 based on the dendrogram in Figure 2 (modify label Figure 4 by Figure 2 in the legend of the Table), which increased the genotypes with clones from eight to 19 and increased the trees with clones from 71 to 83. Consequently, the trees of different genotypes are 105 (86+19). This should certainly modify the results presented in your new 3.2. section.
2) Related to terminology, I suggest using the term clone instead of ramet and avoid the use of genet and ecotype. The terms genet (genetic population) and ramet (suckers or other vegetative propagules from a tree multiplied by seedlings) came from forestry, where seedlings are the usual method of multiplication. Obviously, the adaptation of different genets to an environmental factor, as for instance altitude, determines different ecotypes. In contrast, in the case of the olive, the same circumstances may only happen in wild and feral olives, the latter being escapes by sexual reproduction of previously selected cultivars. As you stated in your response R6, the most common cultivars in the Campania Region (Reference samples) are “vegetative propagated by farmers long time ago”, i.e. there are clones and therefore, the only source of genetic variability must be mutation. Only in the case of feral olives recombination plays a role in increasing biodiversity.
3) In the case of centennial olives, Structure has proved to be an efficient tool to discriminate cultivars, feral and wild olives (See reference 37). I suggest verifying if the admixture index in Figure 2 may be related to the presence of wild or feral olives in your samples. Interestingly, in agreement with your results, most of the centennial olives in reference 37 are paleo cultivars and only few correspond to current catalogued cultivars.
4) You stated in R6 that “Our genetic analysis reveals that cultivar identification is a difficult and tricky job that could confuse the general framework, confirming once again that the real number of olive genotypes is still unknown, and the germplasm comprises accessions/varieties that have not yet been catalogued and deeply studied”. I agree that the number of olive genotypes is still unknown and obviously will be continuously changing, but the number and correct names (true to type) of cultivars is an urgent task for olive growing. In fact, undue association between cultivar denominations and true to type cultivars names is main reason for the chaos evidenced in the successive and exhaustive Olea databases (DOI: 10.7349/OLEA databases) published by Bartolini et al. In contrast the complementary use of SSRs and a limited number of stones’ discrete characters from the UPOV morphological descriptors has proved to efficiently discriminate cultivars in the Olive Council International collections’ in Córdoba (See reference 10) and Marrakech (Khadari et al.
in Olivebiotech 2019). This methodology has evidenced duplicated cultivars in the initial tree accessions and has established homonyms, synonyms, molecular variants, true to type and erroneous denominations due to mislabeling or other circumstances. The resulting list of varietal denominations avoids the previous confusion of the samples planted in those collections.
Therefore, I propose:
1) As stated in your Ms, your main aim is develop an integrated approach based on OliveR software for analyzing biodiversity in olive genotypes, I would accept the Ms for publication once the authors modify Table S5 and text of the Section 3.2. (See consideration 1) accordingly.
2) I kindly suggest the authors to apply the methodology indicated in consideration 4 to their trees, including the Reference samples and the accessions kept in the repository of the Azienda Agricola Improsta of Campania Region.
